# Lung Inflammasome Activation in SARS-CoV-2 Post-Mortem Biopsies

**DOI:** 10.3390/ijms232113033

**Published:** 2022-10-27

**Authors:** Lucas Baena Carstens, Raissa Campos D’amico, Karen Fernandes de Moura, Eduardo Morais de Castro, Flávia Centenaro, Giovanna Silva Barbosa, Guilherme Vieira Cavalcante da Silva, Isadora Brenny, Júlio César Honório D’Agostini, Elisa Carolina Hlatchuk, Sabrina Pissette de Lima, Ana Paula Camargo Martins, Marina De Castro Deus, Carolline Konzen Klein, Ana Paula Kubaski Benevides, Seigo Nagashima, Cleber Machado-Souza, Ricardo A Pinho, Cristina Pellegrino Baena, Lúcia de Noronha

**Affiliations:** 1Laboratory of Experimental Pathology, School of Medicine and Life Sciences, Pontifícia Universidade Católica do Paraná (PUCPR), R. Imaculada Conceição, 1155-Prado Velho, Curitiba 80215-901, PR, Brazil; 2Postgraduate Program of Health Sciences, School of Medicine and Life Sciences, Pontifícia Universidade Católica do Paraná (PUCPR), R. Imaculada Conceição, 1155-Prado Velho, Curitiba 80215-901, PR, Brazil; 3Hospital Marcelino Champagnat, Av. Presidente Affonso Camargo, 1399-Cristo Rei, Curitiba 80050-370, PR, Brazil; 4Postgraduate in Biotechnology Applied in Health of Children and Adolescent, Faculdades Pequeno Príncipe (FPP), Instituto de Pesquisa Pelé Pequeno Príncipe (IPPPP), R. Silva Jardim, 1632-Água Verde, Curitiba 80230-020, PR, Brazil; 5Departmnet of Medical Pathology, Universidade Federal do Paraná (UFPR), Rua General Carneiro, 181-Alto da Glória, Curitiba 80215-901, PR, Brazil

**Keywords:** COVID-19, inflammasome, pyroptosis, cytokine, immunohistochemistry, pulmonary tissue, oxidative stress

## Abstract

The inflammasome complex is a key part of chronic diseases and acute infections, being responsible for cytokine release and cell death mechanism regulation. The SARS-CoV-2 infection is characterized by a dysregulated cytokine release. In this context, the inflammasome complex analysis within SARS-CoV-2 infection may prove beneficial to understand the disease’s mechanisms. Post-mortem minimally invasive autopsies were performed in patients who died from COVID-19 (*n* = 24), and lung samples were compared to a patient control group (*n* = 11) and an Influenza A virus H1N1 subtype group from the 2009 pandemics (*n* = 10). Histological analysis was performed using hematoxylin-eosin staining. Immunohistochemical (IHC) staining was performed using monoclonal antibodies against targets: ACE2, TLR4, NF-κB, NLRP-3 (or NALP), IL-1β, IL-18, ASC, CASP1, CASP9, GSDMD, NOX4, TNF-α. Data obtained from digital analysis underwent appropriate statistical tests. IHC analysis showed biomarkers that indicate inflammasome activation (ACE2; NF-κB; NOX4; ASC) were significantly increased in the COVID-19 group (*p* < 0.05 for all) and biomarkers that indicate cell pyroptosis and inflammasome derived cytokines such as IL-18 (*p* < 0.005) and CASP1 were greatly increased (*p* < 0.0001) even when compared to the H1N1 group. We propose that the SARS-CoV-2 pathogenesis is connected to the inflammasome complex activation. Further studies are still warranted to elucidate the pathophysiology of the disease.

## 1. Introduction

The COVID-19 pathogenesis is linked to a systemic inflammatory reaction with a disproportionate cytokine release and an inability to shift the immune response from innate to adaptive [1]. The disease’s clinical effects are related to an exaggerated immune response characterized by a cytokine storm mediated by the NF-κB pathway [2,3,4] that, in severe cases, leads to pneumonitis, endotheliitis, immunothrombosis, multi-organ dysfunction syndrome (MODS), and ultimately death [5].

Upon entering the body, the SARS-CoV-2 virus binds to the respiratory tract epithelial cells. The angiotensin converting enzyme-2 (ACE2) as a viral receptor will then activate transmembrane protease serine-2 (TMPRSS2) in order to allow the virus to effectively infect the cell [6]. However, literature suggests that the Toll-like receptor 4 (TLR4) could also be activated upon cell membrane interaction with the virus [7]. The NLRP-3 (or NALP) inflammasome complex is a multiprotein complex mediated and activated by both ACE2 and TMPRSS2 [8], and it may have a central role in the disproportionate cytokine release and immunothrombotic COVID-19 repercussions [9]. Studying its activation within COVID-19 cases may provide us with valuable insight into the disproportionate inflammation observed in the disease [10].

The inflammasome is a high molecular weight protein complex activated within the innate immunity reactions cascade that acts as a cell response to infection, intracellular changes, or tissue death [11]. Its activation starts with a first signal recognized by a Pattern Recognition Receptor (PRR) among which we can highlight Toll-like Receptors (TLR) and Nod-like Receptors (NLR) [12]. This complex is regulated by the NF-κB pathway that will in its turn modulate and be modulated by oxidative stress and culminating in the production of interleukin-1β (IL-1β) and interleukin 18 (IL-18) [13]. In addition to the interleukins 1β and 18 activation, the inflammasome complex also leads to a cell death called pyroptosis [14,15].

The inflammasome complex itself is composed of three main components: (1) an NLRP-3 protein, (2) an ASC protein with a CARD domain that will form an (3) oligomer ASC/ NLRP-3. This ASC/ NLRP-3 oligomer activate pro-caspase 1 to caspase-1 (or CASP1), and its assembly is determined by the NF-κB pathway and its nuclear transcription modulation [16].

There are several different inflammasome complexes, each one activated in response to a type of cellular insult. Literature describes inflammasome complexes activated by pathogens such as bacteria, fungi, inorganic molecules, cytokines, and even ionic imbalances originating either intracellularly or extracellularly [16].

The pyroptosis process is characterized by activation of CASP1 (but not by caspase 9 or CASP9 activation) and Gasdermin-D (GSDMD) cleavage resulting in membrane pore formation and cell death [17]. On a large scale, pyroptosis is an important factor of tissue damage, producing cytokines that contribute to Multiple Organ Distress Syndrome (MODS) and often leads to tissue fibrosis and loss of primary function [18].

The inflammatory cascades that lead to the inflammasome pathway activation have a central role in chronic diseases such as atherosclerosis [19], cancer [20], and Crohn’s disease [21]. Moreover, other viral infections such as Dengue fever and Ebola are also linked to pathological inflammasome activation [22,23,24].

The 2009 Influenza A virus H1N1 subtype pandemics (H1N1pdm09) posed several challenges to scientists and healthcare providers in the past decade. The pathogenesis of this specific flu type is characterized by intense interleukin-17 (IL-17) secretion, alveolar inflammatory infiltrate that would impair the lung’s ability to exchange oxygen [25], whereas the SARS-CoV-2 infection and lung tissue damage generates fibrosis and has mast-cell recruitment into pulmonary tissue [26,27]. Comparing both pandemic viruses and their pathogenesis may offer significant insight into relevant differences between their mechanisms.

In this paper, we test a possible mechanism for the SARS-CoV-2 pathology through the observation of inflammasome complex activation markers (TLR4; ACE2; IL-1β; IL-18; NF-κB; ASC; NLRP-3 (or NALP); CASP1; CASP9; GDSM-D; NOX4; TNF-α) in *post-mortem* lung biopsies from patients that died from COVID-19 and compare them to samples from patients that died from non-acute pulmonary death causes and patients that died of the H1N1pdm09.

## 2. Results

### 2.1. Study Sample

The population mean age for patients in the COVID-19 group was significantly higher with a median of 72.5 years when compared to H1N1 (median of 45) and control patients (median of 44) (*p* < 0.0001), COVID-19 patients had a significantly longer time from admission to death (COVID-19× CONTROL *p* = 0.0013/COVID-19 × H1N1 *p* < 0.0005). Other population aspects are described in Table 1.

### 2.2. Immunohistochemistry

ACE2 expression is upregulated by SARS-CoV-2 infection and used by the virus as a way of entering the cytosolic environment [9,28], we have observed that the ACE2 expression was significantly higher (*p* = 0.0001). We have also observed the NOX-4 and NF-κB expression, described by literature as inflammasome upregulators [15], to be significantly increased (*p* < 0.05). IL-1β and IL-18 are direct results of inflammasome activation [28], our analysis showed that they were increased in the COVID-19 group when compared to the control (*p* < 0.0005) group and that IL-18 was increased in the COVID-19 group even when compared to the H1N1 group (*p* < 0.0001). The ASC portion of the inflammasome is responsible for the complex’s role of pro-interleukin and procaspase activation [29,30] our observations show that it was significantly more expressed regardless of group comparison (*p* < 0.0005). Pyroptosis is an inflammasome-dependent process, it is marked by CASP1-mediated GSDM-D cleavage and subsequent membrane pore formation [17,31], we observed that CASP1 was significantly more expressed in the COVID-19 group (*p* < 0.0001), while GSDM-D was significantly less expressed in the COVID-19 group (*p* < 0.0005). CASP9 is an apoptosis mediator and was chosen as a negative control for inflammasome activation [30,32]; it was found to be less expressed in the COVID-19 group when compared to the control group (*p* < 0.0001).

All marker results, as well as their significance and which group showed a higher expression are described in Table 2.

IHC analysis of key inflammasome cytokines and proteins is represented in Figure 1 and Figure 2.

## 3. Discussion

The inflammasome complex proteins and modulators were found to be activated in COVID-19 and its activation was significantly higher when compared to the control patients.

Throughout the analysis of our data, we found that the cell receptors used by the virus (ACE2 and TLR4) were significantly more expressed. We also found that the upregulation of the NF-κB pathway was shifted towards acute inflammasome activation, and this shift became evident upon analyzing the ASC protein expression. Not only we identified an upregulation of ACE2 and NF-κB, which are essential for the inflammasome complex activation, but also we observed the increased expression of IL-1β and IL-18, which are some of the cellular products of acute inflammasome activation.

We observed that the patients suffering from COVID-19 underwent a process of cell death called pyroptosis, which is dependent on inflammasome activation and shows a very specific caspase expression, relying on CASP1 (but not CASP9). This finding is relevant because even though chronic illnesses are known to activate the inflammasome complex, the shift towards pyroptosis indicates a disproportionate acute activation determinant to the type of tissue lesion and subsequent fibrosis [27,33], whereas CASP9 is seen mainly as a late response to hypoxic insult and in physiological apoptosis [32,34].

### 3.1. Population

Regarding the population analyzed, we observed that the COVID-19 group was significantly older when compared to any of the other groups.

Literature describes age as one of the key aspects of an oxidative stress imbalance called inflammaging in which the immune system’s ability to shift from innate to adaptive immunity is impaired [17]. However, analyzing the main causes of death within the control group, several chronic inflammatory diseases were described as cancerous diseases that are especially relevant when analyzing the inflammasome activation [31,32].

The demographic information ensemble shows us that both the control group, and the COVID-19 group, suffered from chronic inflammatory diseases. These demographical differences were not enough to compensate the inflammasome activation observed on the COVID-19 group.

The control group samples serve as an approximation to normality but were not disease free, their medical histories describe chronic illnesses such as hypertension, diabetes, and malignancies that are hallmarks for chronic inflammasome activation [33] but did not include any concomitant bacterial infections. The H1N1 group served as a positive control group to show that other pandemics acute infectious diseases have a different inflammatory profile.

### 3.2. H1N1 Death Process

The H1N1 pathogenesis is characterized by a secretion of interleukin-17 (IL17) that recruits neutrophils causing extensive edema and lung infiltrate. Its infection engenders extensive neutrophilic and macrophagic infiltrate into the alveolar sacs impairing the patient’s ability to properly exchange oxygen with the air. The production of IL-17 does, however, upregulate the production of IL-1β, IL-18, and IL-8, and to some degree upregulates the NF-κB pathway [34], this upregulation does not translate, however, into the pyroptosis process but rather on a Caspase-3 induced apoptosis [35,36]. In this paper, we chose H1N1 to be a positive control for an inflammatory disease that does not involve NLRP3-induced pyroptosis.

### 3.3. The Inflammasome Complex

Activation of inflammasomes is different in chronic diseases and acute diseases, chronic diseases rely on different pathways in action such as the canonical, non-canonical, and alternative activation pathways [32,37]. Viral infections tend to regulate the inflammasome activation through the canonical activation pathway. We observed that the SARS-CoV-2 infection causes a disproportionate activation of the canonical inflammasome pathway [38], which is illustrated in Figure 3.

Our results also show that the ACE2 expression was significantly higher in COVID-19 when compared to both H1N1 and Control groups, as seen on Figure 1. This difference could be attributed to the fact that the SARS-CoV-2 is known to not only use ACE2 as a way to infect a cell, but also to upregulate its expression as a result of its infection [39]. Another possible mechanism increasing the ACE2 expression is that age is a constitutive factor to overexpressing ACE2 on cell membrane [40].

TLRs are pathogen binding cell membrane receptors that are widely present in respiratory tract epithelial cells and are key immune response regulators acting as a bridge between native and innate immune responses [12,25]. Once the TLR4 receptor binds to an antigen, a shift in cell metabolism is triggered and the NF-κB pathway is activated. Upon infection, mitochondrial stress is increased and an imbalance occurs leading to the expression of mitochondrial NADPH oxidase 4 (NOX4) [41]. The presence of NOX4 in the cytosolic environment, as well as oxidative stress species and ionic changes created by mitochondrial dysfunction, modulates the production of pro-caspases and pro-interleukins, as well as NLRP-3 oligomers that may go on to polymerize and form an inflammasome complex polymer [11]. Our study shows that the expression of NOX4 was significantly higher in COVID-19 patients, regardless of which group comparison was made, as seen on Figure 1, demonstrating that the SARS-CoV-2 infection leads to important oxidative stress and mitochondrial imbalance.

The NF-κB pathway is a multi-reaction metabolic pathway that can either determine cell survival or cell death [42]. Upon cell infection, there is a shift from maintaining cell homeostasis and survival to an inflammatory response [43] that is mainly composed of IL-1β [43,44]. In this context, observing the NF-κB expression gives us valuable insight on the effects the SARS-CoV-2 has on the cellular machinery. We observed that the COVID-19 group presented the highest NF-κB expression within pulmonary tissue, as seen on Figure 1. This finding may be a clue to one of the many pathogenesis differences between H1N1pdm09 and COVID-19, as well as an indicator of the tissue reactions observed.

This study shows that the NF-κB protein was significantly increased further corroborating that the inflammasome activation is plausible and likely. Not only was the NF-κB expression increased but also the interleukin profile identified upon the biopsies shows that the pathway modulated by the NF-κB was indeed shifted towards an inflammatory response relying on the inflammasome activation [45].

The ASC protein is a part of the inflammasome complex; more specifically, it is the part in which the activation of pro-interleukins and pro-caspases takes place. NLRP-3 oligomers and polymers within the inflammasome complex rely on the CARD domain within the ASC protein to activate IL-1β, IL-18, and CASP1 in the cytosolic environment [46]. The increased expression of ASC is indicative that not only cell metabolism has shifted to the NLRP-3 inflammasome but also that it is indeed capable of producing the cytokines that will further modulate cell death and injury response [16,46]. Our study showed that, although there was no difference in the NLRP-3 component expression the COVID-19 group had a significantly higher ASC expression, as seen on Figure 1, which indicates inflammasome activity.

IL-1β and IL-18 are interleukins belonging to the IL-1 family [47] and are produced as a response to cellular insults that range from infection, such as the classically described LPS-induced inflammasome activation, to metabolic imbalances resulting in ionic fluxes through cell membrane [8,48,49]. IL-1β is well described as especially important in chronic metabolic diseases [11,21,33] and in acute infectious diseases, among which we can highlight Dengue virus, Mayaro virus, and even Ebola virus [23,50]. It is important to note that H1N1pdm09 will also cause an increase on IL-1β expression but what is seen is that its expression is due to other inflammatory pathways as well as, on a minor scale, the inflammasome. The IL-1β plays an important role as a second signal in the inflammasome activation. In macrophages, it acts as an upregulator to pathogen phagocytosis, in the endothelium, it will modulate further inflammatory cytokine secretion and increase the expression of adhesion molecules involved in macrophage recruitment [51]. Its role, however, is extended to the extracellular level as it acts as an activator to immune cell-secreted cytokines and proteases [47]. Both IL-1β and IL-18 act dually as a product of NLRP-3 activation and regulators of its effect [16,52], IL-1β expression is represented on Figure 2.

Taken together, the increased expression of ACE2, NOX4, NF-κB, and ASC indicate that severe cases of COVID-19 suffered from an increased inflammasome complex activation even when compared to H1N1 and to a control group with chronic inflammatory illnesses. Adding to that evidence the increased expression of both IL-1β and IL-18, as inflammasome activation products and as inflammasome complex upregulators, corroborate our hypothesis of its activation in COVID-19.

### 3.4. Pyroptosis

CASP1 is a pyroptotic caspase produced in the form of pro-caspase 1, it serves as an effector to cell death in response to inflammatory injury. The modulation of its production is NF-κB pathway-dependent, determines membrane pore formation, and increases the maturation of IL-1 family interleukins such as IL-1β and IL-18. Inflammasome-activated pyroptosis is dependent on CASP1, in our study we demonstrated that CASP1 was expressed and that it was significantly higher when compared to both H1N1 and control groups, as seen on Figure 2. Pyroptosis is an inflammatory cell death mechanism activated by the NLRP-3 inflammasome complex, mediated by CASP1. It differentiates itself from apoptosis and other programmed cell death processes by forming a characteristic membrane pore and by not being dependent on CASP9 [53,54].

CASP9 is an effector caspase determinant to a shift into apoptosis, this cytokine is not only relevant within the embryological development in which it regulates certain tissue regressions but is also important when regulating certain responses to injuries such as hypoxia. Although this protein also serves as an initiator for cell death its activation leads to apoptosis cell death which keeps the membrane intact [32,55,56]. Literature describes CASP9 as an antagonist to the pyroptosis process observed in our COVID-19 group [30,32,57]. CASP9 expression is seen on Figure 2.

Morphologically pyroptotic cells present themselves with DNA fragmentation but maintain the nucleus intact; its main characteristic is cell membrane integrity failure due to its membrane pores and consequent osmotic lysis. Interestingly pyroptosis shares some characteristics with Apoptosis but also with some programmed cell death processes [56].

Once the NLRP-3 inflammasome complex activates pro-caspase 1 into CASP1 the caspase activates GSDM-D, which is in its turn cleaved into GSDM-NT and GSDM-CT in order to dimerize itself and form the membrane pore that characterizes pyroptosis [56]. GSDM-D expression is represented on Figure 2.

The ensemble of all markers expressed leads us to believe that not only the inflammasome complex is activated, since there is a higher IL-1β expression but also a GSDM-D cleavage that indicates that it is being polymerized into membrane pores and its regulating caspases, that indicate the pyroptosis as being the process for lung injury in COVID-19 [30,54].

### 3.5. Study Limitations

This study analyzed post-mortem biopsies through immunohistochemistry, and had a limited sample size. This kind of analysis is limited to the frame in time in which patients died, and therefore interpretation from data in this paper should take into account the population analyzed and the fact that the samples were obtained post-mortem and hence do not represent the patient’s longitudinal clinical and pathological evolution but rather a frozen frame in time.

Immunohistochemistry studies have limitations as to reproducibility and sensitivity, but our lab has extensive experience in this type of study. Western-Blotting and PCR tests cannot be performed in formalin fixed and paraffin embedded samples, due to the limited access to COVID-19 samples at the beginning of the pandemics, we were not able to further analyze our samples using these techniques [27].

This study findings must also be interpreted with caution since even though our control group suffered from diseases that are hallmarks to chronic inflammasome activation, the higher age of our COVID-19 group may be a bias to our findings.

The strengths of this study are fully showing the inflammasome pathway, from the cellular receptors and the first signal to the NF-κB protein expression, after that we showed inflammasome activity by measuring not only its resulting cytokines but also by showing the cell’s metabolical shift towards pyroptosis.

## 4. Materials and Methods

### 4.1. Ethics Committee Approval

#### Samples

Clinical data and post-mortem lung biopsy samples were obtained from twenty-four patients that died from COVID-19 at the Intensive care unit (ICU) in Hospital Marcelino Champagnat in Curitiba-Brazil. Samples were then formalin-fixed and paraffin-embedded (FFPE).

A control group of eleven FFPE post-mortem lung samples was collected at Hospital de Clínicas, Curitiba-Brazil composed of ten patients, the major causes of death within this group were cancer (gastric, hepatic, laryngeal, neuroendocrine carcinoma, and lymphoma), acute myocardial infarction, peritonitis, and dementia/cachexia.

Samples from FFPE post-mortem lung biopsies of patients that died from the Influenza A virus H1N1 subtype pandemic virus in 2009 (H1N1pdm09) were obtained from patients treated at Hospital de Clínicas.

Patients from both COVID-19 and H1N1 groups had their infection confirmed through Real-Time Polymerase Chain Reaction (qRT-PCR) tests. Samples from patients in the Control group were obtained from patients that died before the COVID-19 pandemics.

### 4.2. Immunohistochemical Analysis

FFPE samples were obtained and fixed, after that, they underwent staining with hematoxylin and eosin-H&E (Harris Hematoxylin: NewProv, Cod. PA203, Pinhais, Brazil; Eosin: BIOTEC Reagentes Analíticos, Cod. 4371, Pinhais, Brazil). Subsequently, specific staining for TLR4; ACE2; IL-1β; IL-18; NF-κB; ASC; NLRP-3 (or NALP); CASP1; CASP9; GDSM-D; NOX4; TNF-α was performed, and the slides were then scanned using Axio Scan.Z1 Scanner (ZEISS, Jena, Germany), and then ZEN Blue Edition (ZEISS, Jena, Germany) was utilized to randomly generate high-power fields (HPF  =  40× objective). Images were randomly generated by the software, with no investigator’s interference. The immunopositivity areas were measured by the Image-Pro Plus software version 4.5 (Media Cybernetics, Rockville, MD, USA). Subsequently, these stained areas were converted into percentages per total tissue area to enable statistical analysis. This study analyzed overall marker expression. Staining information can be found on Appendix A

### 4.3. Statistical Analysis

Statistical analysis was done using GraphPad Prism software version 9.4.0, San Diego, CA, USA. The values obtained from the clinical data and sample digital analysis were tested for normality and were then analyzed through the appropriate test. The tests used were Student’s *t*-test and the Mann–Whitney test when appropriate. Categorical variables such as gender underwent a Chi-square and Fisher’s exact tests when appropriate.

## 5. Conclusions

In conclusion, we demonstrated that the SARS-CoV-2 virus uses cell entry mechanisms that cause NLRP-3 inflammasome activation, as well as Inflammasome activity and pyroptosis as a consequence of its activation covering the beginning of the process with cell infection, second signal with mitochondrial dysregulation NF-κB pathway modulation up until its final products as IL-1β and IL-18 as well as CASP1, we also differentiated the pyroptotic cell death process from other processes by measuring GDSMD and CASP9. We found that the inflammasome complex is highly activated in patients that died from COVID-19 and that it could be an important part of the pathogenesis of the disease. Longitudinal studies are still warranted to establish that the inflammasome complex is responsible for COVID-19 gravity and morbidity. Our findings are relevant for understanding how the virus affects the cell upon severe infection, they may help build an understanding of the core mechanisms targeted within SARS-CoV-2 therapeutics and new viral variations effects and changes.

## Figures and Tables

**Figure 1 ijms-23-13033-f001:**
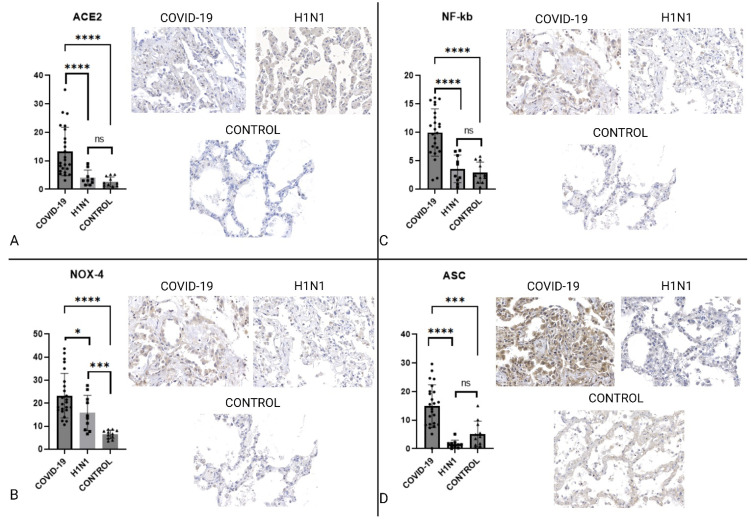
Graphs showing the comparison between COVID-19, H1N1, and CONTROL groups concerning ACE2, NOX4, NF-κB, and ASC. Panel (**A**) shows that ACE2 is significantly increased in COVID-19 when compared to H1N1 and the control group. In Panel (**B**), NOX4 is more expressed in COVID-19 than in the H1N1 and CONTROL groups. Images (**C**) and (**D**) also show that NF-κB. and ASC expression are remarkably high in COVID-19. These graphs show that the COVID-19 expressed characteristics needed for SARS-CoV-2 cell entry as well as oxidative stress and key components for inflammasome activity. The symbol “ * ” stands for *p ≤* 0.05, “ *** “ stands for *p* ≤ 0.001, “****” stands for *p ≤* 0.0001. Slide images were obtained on a 40× augmentation. Image created using biorender.com.

**Figure 2 ijms-23-13033-f002:**
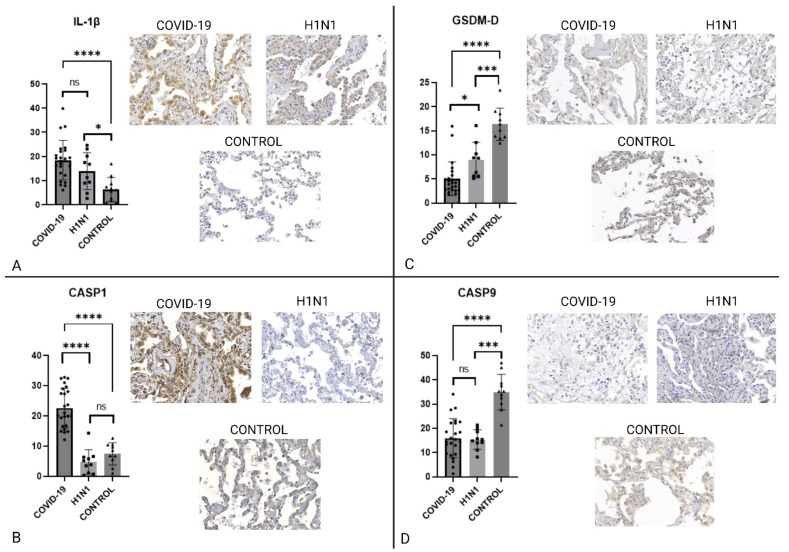
Graphs presenting the comparison between COVID-19, H1N1, and CONTROL groups regarding IL-1β, CASP1, GSDMD, and CASP9. Panel (**A**) shows that IL-1β is more expressed in COVID-19 than in H1N1 and the control group. In Panel (**B**) is possible to see that CASP1 expression is remarkably high in COVID-19, unlike CASP9 (seen in Panel (**D**)) which is significantly less expressed, a difference expected to be seen only within the Pyroptosis context. Panel (**C**) shows that GSDMD is being less expressed in COVID-19 than in H1N1 and CONTROL groups. The symbol “*” stands for *p* ≤ 0.05, “***“ stands for *p* ≤ 0.001, “****” stands for *p* ≤ 0.0001. Slide images were obtained on a 40× augmentation. Image created using biorender.com.

**Figure 3 ijms-23-13033-f003:**
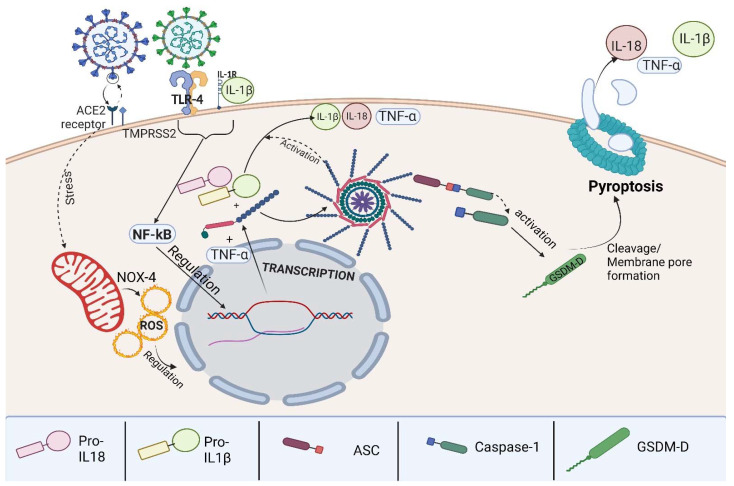
Graphical abstract demonstrating the viral infection stimulating ACE2 receptors and TLR4 receptors, modulating oxidative stress, and stimulating the NF-κB pathway followed by protein transcription culminating in pro-IL-1β, pro-IL-18, and TNF-α production. The process is then followed by NLRP-3 activation of pro-IL-1β, pro-IL-18, CASP1 activation, and GSDM-D cleavage leading to pore membrane formation.

**Table 1 ijms-23-13033-t001:** Sample demographic data.

Characteristics	COVID-19	H1N1	Control	*p*-Value
Female	9 (37.5%)	2 (20%)	3 (27.27%)	NS
Male	15 (62.5%)	8 (80%)	8 (72.73%)	
Age (median in years)	72.5	44	45	COVID-19 vs. H1N1 *p* < 0.0001 COVID-19 vs. CONTROL *p* < 0.0001
Time from admission to death (median in days)	13.0	1.5	4	COVID-19 vs. H1N1 *p* < 0.0005 COVID-19 vs. CONTROL *p* = 0.0013
Duration of invasive ventilation (median in days)	9.5	1.5	N/A	*p* = 0.0096
Death Cause	Diffuse Alveolar Damage and Disseminated Coagulopathy	Diffuse Alveolar Damage	Peritonitis, Infarction (*n* = 3), Neuroendocrine Carcinoma, Adenocarcinoma, Hepatic Cancer, Laryngeal Cancer, Surgical Complications, Lymphoma, Thrombosis	N/A
Comorbidities	Hypertension (*n* = 21), Chronic Cardiac disease (*n* = 11),	Data Not Obtained	Hypertension (*n* = 3), Chronic Cardiac disease (*n* = 5),	
	Malignancy (*n* = 3)		Malignancy (*n* = 5)	
	Diabetes Mellitus type 2 (*n* = 11)		Diabetes Mellitus type 2 (*n* = 2)	
	Dyslipidemia (*n* = 17)		Dyslipidemia (*n* = 4)	
	Obesity (*n* = 6)		Obesity (*n* = 4)	
	Chronic Lung disease (*n* = 5)		Chronic Lung disease (*n* = 4)	

**Table 2 ijms-23-13033-t002:** Results of statistical analysis and immunohistochemical analysis. Column 1 shows the staining analyzed, Column 2 shows the COVID-19 group versus H1N1 comparison, and Column 3 shows the COVID-19 versus CONTROL comparison. Paired marker expression comparison. Arrows pointing upwards indicate augmented expression.

Marker	COVID-19 × H1N1	*p*-Value	COVID-19 × Control	*p*-Value
ACE2	↑COVID-19	0.0001	↑COVID-19	<0.0001
TLR4	↑H1N1	0.0247	↑CONTROL	0.0164
NLRP-3/NALP	NS	0.4615	NS	0.1628
IL-1β	NS	0.1439	↑COVID-19	<0.0001
IL-18	↑COVID-19	<0.0001	↑COVID-19	0.0004
NF-κB	↑COVID-19	<0.0001	↑COVID-19	<0.0001
ASC	↑COVID-19	<0.0001	↑COVID-19	0.0004
CASP1	↑COVID-19	<0.0001	↑COVID-19	<0.0001
CASP9	NS	0.8332	↑CONTROL	<0.0001
GSDMD	↑H1N1	0.0003	↑CONTROL	<0.0001
NOX4	↑COVID-19	0.0372	↑COVID-19	<0.0001
TNF-α	NS	0.0929	↑COVID-19	0.0011

## Data Availability

Not applicable.

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
