# Peer review of "Lung Inflammasome Activation in SARS-CoV-2 Post-Mortem Biopsies"

_ijms, 2022, doi:10.3390/ijms232113033_

Round 1

Reviewer 1 Report

Overall, this is an interesting study in which the authors investigated the contribution of inflammasome complex activation that contributes to the pathogenesis of COVID-19.

 Major comments:

1-     The authors concluded that “inflammasome complex is highly activated in patients that died from COVID-19, and that could be an important part of the pathogenesis of the disease.” This conclusion was based on the results observed from the immunohistochemistry. Despite the wide use of immunohistochemistry in research, immunohistochemistry is considered a semi-quantitative technique as it is subject to variability in sensitivity and reproducibility. Therefore, other techniques such as PCR or western blotting are required to support the study's findings and strengthen the manuscript.  

2-     In the current study, the population mean age for patients who died from COVID is significantly higher than the control patients. Several studies have shown that aging is associated with increased production of inflammatory markers and the activation of NLRP3 “Inflammaging.” Therefore, the reported increase in inflammasome complex proteins and modulators in the COVID group can be the result of aging, not just COVID infection.

3-     Several diseases, including but not limited to COPD, have been associated with an increase in inflammatory response and inflammasome activation. More information about the patient’s medical history is needed to strengthen the manuscript.

Author Response

Please see attatchment.

Reviewer 2 Report

In this manuscript the authors evaluated the activation of NLRP3 inflammasome in SARS-CoV-2 patients, by performing IHC analysis on post-mortem lung specimens. the manuscript is overall well written, and the message clear anyhow I have few suggestions.First of all I would modify the results and discussion sections, implementing the results description including some explanations and leave to the discussion only the speculative part. In fact, in the present form there is an unbalance between these two sections, that in my opinion reduces the importance of the presented data. 

Secondly, in order to facilitate the comprehension, the p values should be represented as asterisks, following the well known scientific rules. The table 2 resuming the results can be added as supplementary, while major impact should be reserved to the graphs and images. These figures should be moved in the results sections and before the graphical abstract, that if it has to be present in the publication should be either in the front page next to the abstract, or even better at the end of the discussion, as a simplification or resume of the presented data.

As concerned the representative images, they should be bigger, with insets that allow the reader to see the different staining intensity. ACE2 representative images are not so convincing, compared to the data showed in the plot, as well as GSDM-D and CASP-9. Please consider to find alternative representative images. The plots should all display also the mean +- either SEM or SD.

Minor:

Line 127-129: the concept is not celar.

Lines 146-148: please rephrase.

Lines 158-161: Here it is not clear what is the message. Please explain and rephrase.

Author Response

Please see attatchment.

Reviewer 3 Report

The authors performed a post-mortem cytokine IHC analysis on FFPE samples from human COVID-19 patients, H1N1 (2009 pandemic) patients and control patients that dies of other causes, in attempt to test whether inflammasome activation is involved in the gravity of symptoms and mortality in COVID-19 patients.

Comments:

1) Post-mortem analysis of secreted cytokines may be tricky and probably not very clearly determined via IHC. Also not clear if patient history or additional (bacterial) infections that usually complicate lung infections was obtained/taken into account during the analysis. The obtaining and the quality of the samples is not discussed. The Authors mention that some of the control patients that died from cancer may have cancer-related inflammation processes that could have been captured and thus bias the comparisons.

2) The authors mention that the inflammasome was significantly more activated in the COVID19 patients (lines 122-123) compared to both the controls and the H1N1 patients, but in Table 2 the inflammasome proteins (NLRP-3/NALP) were not significantly higher in the comparisons with either H1N1 or control groups. Also IL-1b expression was not significantly higher compared to the H1N1 group.

3) Could the authors address the different mechanisms of H1N1- and COVID19- induced/related death?

4) Additionally as described in the introduction and in figure 1, gasdermin pore formation is important for IL-1b and IL-18 secretion/signalling and the further cytokine storm responses, so it should be higher in COVID19 patients (?), yet the Control samples had highest gasdermin expression (Table 2 and Figure 3). Could the authors comment on that?

5) In the materials and methods section the authors should describe a little more the pipeline/analysis of the samples. Also they mention the areas were converted into percentages, but not mentioned percentages of what? Intensity of staining? Area size of field? Please elaborate.

6) Did the used antibodies take into account the "activation" status of the inflammasome components, or only their overall expression?

7) Please provide a reference for statement on lines 136-137.

8) Some minor grammar and typo mistakes caught:
- Line 37: Should be "Data obtained from digital analysis underwent appropriate statistical analysis."
- Line 56: ACE2 and TMPRSS2 (as well as any other abbreviations in the manuscript) should be written in full the first place of mention.
- Lines 57-58: double "also" in the sentence. Please remove one.
- Line 74: Should be "...pro-caspase 1 to caspase-1..."
- Line 91: A new sentence should start after "...in the past decade..."
- Line 95: Should be "...both pandemic viruses..."

Round 2

Reviewer 1 Report

No further comments

Author Response

Thank you for your comments

Reviewer 2 Report

I would like the authors for the work performed on the manuscript. However, there is still one point that has not been addressed and in my opinion is quite important for the overall comprehension of the text:

 "First of all I would modify the results and discussion sections, implementing the results description including some explanations and leave to the discussion only the speculative part. In fact, in the present form there is an unbalance between these two sections, that in my opinion reduces the importance of the presented data. "

The authors replied top this comment explaining that the population had a normal distribution. I think they might got confused.
